# Improving Neuron-level Interpretability with White-box Language Models

## Abstract

Neurons in auto-regressive language models like GPT-2 can be interpreted by ana-
lyzing their activation patterns. Recent studies have shown that techniques such as
dictionary learning, a form of post-hoc sparse coding, enhance this neuron-level
interpretability. In our research, we are driven by the goal to fundamentally im-
prove neural network interpretability by embedding sparse coding directly within
the model architecture, rather than applying it as an afterthought. In our study, we
introduce a white-box transformer-like architecture named Coding RAte Trans-
formEr (CRATE), explicitly engineered to capture sparse, low-dimensional struc-
tures within data distributions. Our comprehensive experiments showcase sig-
nificant improvements (up to 106% relative improvement) in neuron-level inter-
pretability across a variety of evaluation metrics. Detailed investigations confirm
that this enhanced interpretability is steady across different layers irrespective of
the model size, underlining CRATE's robust performance in enhancing neural net-
work interpretability. Further analysis shows that CRATE's increased interpretabil-
ity comes from its enhanced ability to consistently and distinctively activate on
relevant tokens. These findings point towards a promising direction for creating
white-box foundation models that excel in neuron-level interpretation.

## 1 Introduction

*Representation learning* aims to learn a continuous mapping, to transform a random vector in a high
dimensional space that is sampled from a dataset, to a feature vector in another (typically lower-
dimensional) space (Bengio et al., 2013). Recently, deep learning has witnessed tremendous empir-
ical success in modeling massive amounts of high-dimensional data, and the predominant practice
has been to learn first a task-agnostic representation by pre-training a large neural network, which
is commonly known as the *foundation model* (Devlin et al., 2019; Radford et al., 2019). Among
language foundation models, the transformers architecture (Vaswani et al., 2017) with Generative
Pre-Training (Radford et al., 2019) (GPT) has recently demonstrated a strong capability of model-
ing sequential data and thus predicting subsequent tokens (Brown et al., 2020; Ouyang et al., 2022).
Such strong capability has emerged significant success in downstream applications (Lewis et al.,
2020; Yang et al., 2023), yet the large neural network is known to be *black-box*, where the represen-
tations in the model are not independently interpretable, introducing difficulty in designing effective
paradigms for major known challenges of (visual) language models like hallucination (Ji et al.,
2023; Tong et al., 2024), bias (Garrido-Muñoz et al., 2021; Nadeem et al., 2020), and catastrophic
forgetting (Kemker et al., 2018; Zhai et al., 2024).

To interpret the functions of individual modules in the language models, *mechanistic interpretation*
was proposed to reverse-engineer such models, through identifying meaningful patterns in the data
representations and computational mechanisms of the model components (Olah, 2022; Meng et al.,
2022a). Recent studies on auto-regressive models like GPT-2 have delved into *neuron-level inter-
pretation*, where the focus is on understanding the activations within the model's MLP layers (Yun
et al., 2021). This approach helps to reveal the specific roles of individual neurons, which is cru-
cial for precise model editing and control (Meng et al., 2022a;b). Recent research has also proposed
that *sparse auto-encoder* (SAE)-based dictionary learning effectively promotes mono-semanticity of
neurons, thus enhancing neuron-level interpretability (Bricken et al., 2023). However, as a post-hoc
method, sparse auto-encoders always introduces a non-negative reconstruction loss, which intro-
duces noise for steering the model and imperfect fidelity when interpreting the neurons (Bricken

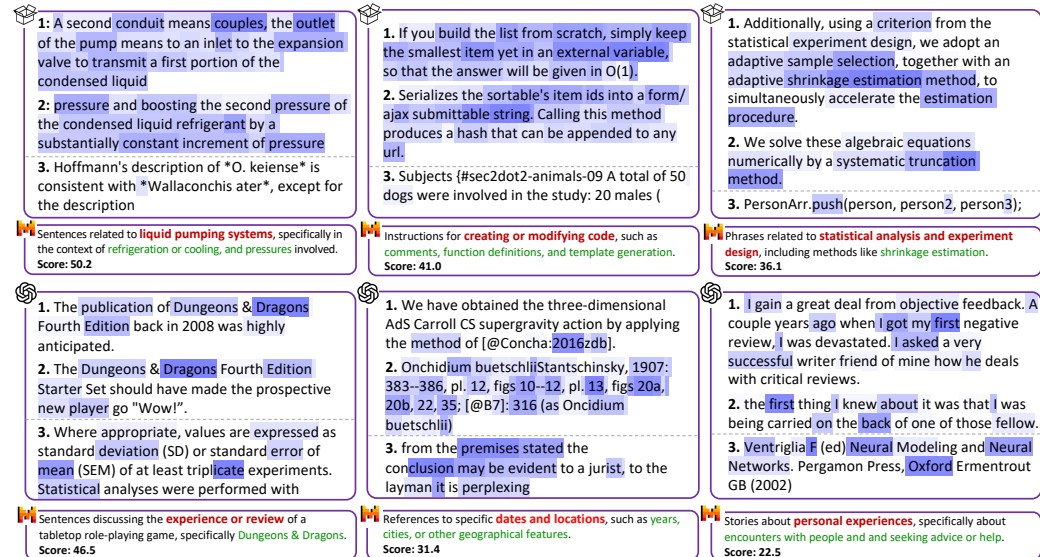

Figure 1: Instances are systematically identified where the interpretability of CRATE (ours, *row 1*) outperforms GPT-2 (*row 2*). For each neuron (*rounded box*), we show two top activated text excerpts (*excerpt 1 and 2*) and one randomly activated excerpt (*excerpt 3*). Results show that CRATE consistently activates *on and only on* semantically relevant text excerpts (first two excerpts), leading to more precise explanations predicted by agents like Mistral.

et al., 2023). More recent studies also show that sparse auto-encoders are hard to scale up, as there exists a significant amount of directions in a neuron in larger language models, which makes the decomposition difficult (Kissane, 2024; Templeton, 2024; Rajamanoharan et al., 2024).

***Can we instead build sparsity directly into the language model?*** In this paper, we develop the CRATE language model, a GPT-2-size language model that builds sparse coding into the model with a mathematically principled way. CRATE handles the problems introduced by sparse encoders at scale: it **(i)** escapes the loss introduced in reconstructing the language model representations, enabling loss-free steering, and **(ii)** escapes the unsteady process of training a sparse auto-encoder. To avoid adding inconsistency into the language model, we develop on top of a mathematically principled white-box model framework, named CRATE (Yu et al., 2023a).[1] After encoding the text tokens into numbers, we apply language-domain-specific modifications to the original CRATE architecture and obtain the token representations. The final representations are then used to predict the next token, while the intermediate representations gets interpreted.

To this end, the main contribution of this work is to propose a causal language model architecture based on the CRATE model framework that builds sparsity inherently, that achieves significantly better neuron-level interpretability (**106% relative increase**) than language models with the GPT architecture under a similar configuration. CRATE forms a family of models from single-layer model up to a 12-layer configuration. Comparative qualitative analysis of neuron activations between CRATE and GPT-2 is provided in Figure 1, alongside extensive quantitative evaluations demonstrating that *by explicitly integrating sparse coding into the language model,* CRATE *achieves markedly improved interpretability across layers compared to GPT-2, applicable across a wide range of model sizes under a variety of evaluation metrics.*

## 2 RELATED WORK

**Neuron-level Interpretation.** Recent studies have provided insights into how auto-regressive models like GPT-2 work at the level of individual neurons, named *neuron-level interpretation*. These studies focus on analyzing *activations*, which are the outputs from the activation functions within the model's multi-layer perceptron (MLP) layers (Yun et al., 2021). Analyzing activations helps

---

[1]In the remaining parts of the paper, we use "CRATE" or "CRATE language model" to refer to our language model architecture, while "original CRATE" denotes the architecture framework described in the literature.

uncover the roles of language model components, which is significant to applications like precise modifications and control on the model, known as model editing (Meng et al., 2022a;b). Neuron-level interpretation is crucial to understanding the mechanisms in a model, including what concepts are learned in the neurons of the network, whether specific neurons are learning particular concepts, and how localized/distributed and redundantly the knowledge is preserved within neurons of the network. A higher neuron-level interpretability indicates that more neurons are interpretable or neurons are more interpretable (Sajjad et al., 2022). As interpretations of the neurons can help localize the knowledge obtained in a neural network, neuron-level interpretation can be used for editing the knowledge in models (Meng et al., 2022a;b), model pruning and distillation (Belinkov et al., 2020), adapting the model to different domains and steering the output (Erhan et al., 2009; Rimsky et al., 2023), and debugging model errors (Hernandez et al., 2021). Improved neuron-level interpretability increases reliability and performance in the applications above.

**Sparse Auto-encoders.** To enhance interpretability, post-hoc sparse coding methods like dictionary learning (Kreutz-Delgado et al., 2003) are used, but these techniques result in imperfect reconstructions and thus always introduces loss when steering the model (Conmy, 2023; Bricken et al., 2023). Literature also indicates that sparse-autoencoders are hard to scale, i.e., a dramatic drop in interpretable features can be observed when models becomes deeper (Kissane, 2024). Additionally, tuning SAE models for larger $L$ values involves extensive hyperparameter tuning and time-consuming training, requiring multiple metrics (reconstruction rate, L1 loss, number of dead neurons) for reliable judgment, which can't be easily optimized with automatic engineering tricks (Bricken et al., 2023).

**Evaluation of neuron-level interpretability.** Metrics now exist to evaluate neuron-level interpretability in language models, examing if neurons trigger on relevant tokens in given contexts (Bills et al., 2023; Bricken et al., 2023). Recent works have demonstrated that a small number of circuits in language models are interpretable (Wang et al., 2022; Chughtai et al., 2023), but comprehending each neuron, out of millions, is vital for thorough model safety audits. Given the prohibitive cost of human evaluation on such a scale, OpenAI introduced an automated metric using large language models for interpretability assessment (Bills et al., 2023), which Anthropic later refined for sparse activations (Bricken et al., 2023). These methods align closely with human judgment and have gained broad acceptance within the research community (Conmy et al., 2024; Liu et al., 2023; Burns et al., 2023; Lieberum et al., 2023). These metrics show that neuron-level interpretability in auto-regressive models is limited (Sajjad et al., 2022), where the popular hypothesis is that neurons are superpositions of simpler semantics, which makes them *fire* (produce a high activation) at multiple semantically distinct sets of tokens (Elhage et al., 2022).

**White-box models and structured representation learning.** In the domain of structured representation learning, *white-box* models stand out for their ability to generate explicit, structured data representations that adhere to specific, desirable configurations such as sparsity and piece-wise linearity, as discussed by Gregor and LeCun (2010) and Chan et al. (2022). Within this framework, Yu et al. (2023a) introduced an innovative approach to constructing deep networks based on unrolled optimization. Specifically, Yu et al. (2023a) proposed the CRATE model, utilizing an information-theoretic objective aimed at promoting the *compression and sparsity* of data towards a predefined statistical structure. Recently, empirical experiments suggest that the white-box design of CRATE inherently develops segmentation capabilities from the data representations at both holistic and component levels with supervised training in the vision domain (Yu et al., 2023b), which directly motivates us to further explore the data representations within such architecture for language models. Recent work has also shown that the CRATE framework is scalable: it can be effectively scaled up to comparable performance as Vision Transformer (ViT) with careful engineering (Yang et al., 2024). Furthermore, the fine-tuning performance of the pretrained CRATE model is also proven to be comparable in both the language domain (Yu et al., 2023a) and vision domain (Yang et al., 2024).

## 3 PRELIMINARIES

This section introduces the original CRATE architecture introduced in Yu et al. (2023a).

**Notations.** In this paper, we denote the one-hot input tokens by $\boldsymbol{X} = [\boldsymbol{x}_1, \ldots, \boldsymbol{x}_N] \in \mathbb{R}^{V \times N}$, where $\boldsymbol{x}_i \in \mathbb{R}^{V \times 1}$ represents the $i$-th one-hot token, $N$ is the total number of input tokens, and $V$ is the vocabulary size. We use $f \in \mathcal{F} : \mathbb{R}^{V \times N} \to \mathbb{R}^{d \times N}$ to denote the mapping induced by the model, which is a composition of $L + 1$ operators (layers) $f = f^L \circ \cdots f^\ell \circ \cdots f^1 \circ f^{\text{pre}}$,

where $f^\ell : \mathbb{R}^{d \times N} \to \mathbb{R}^{d \times N} (1 \leq \ell \leq L)$ represents the mapping of the $\ell$-th operator, and $f^{\mathrm{pre}} :$ $\boldsymbol{X} \in \mathbb{R}^{V \times N} \to \boldsymbol{Z}^1 \in \mathbb{R}^{d \times N}$ represents the pre-processing layer that transforms the one-hot token representations $\boldsymbol{X} = [\boldsymbol{x}_1, \ldots, \boldsymbol{x}_N]$ to semantic embeddings $\boldsymbol{Z}^1 = [\boldsymbol{z}_1^1, \ldots, \boldsymbol{z}_N^1]$. We let $\boldsymbol{Z}^\ell$ denote the input token representations of the $\ell$-th operator $f^\ell$ for $1 \leq \ell \leq L$, so that $\boldsymbol{z}_i^\ell \in \mathbb{R}^d$ denotes the representation of the $i$-th token $\boldsymbol{x}_i$ before the $\ell$-th layer. We denote $\boldsymbol{Z} = \boldsymbol{Z}^{L+1}$ as the output token representations of the last ($L$-th) layer.

**Framework, objective, and optimization.** The transformation of input data into *parsimonious* (piecewise linearized and compact) representations is accomplished by adopting a local signal model for the marginal distribution of the tokens $\boldsymbol{z}_i$. This statement suggests that the tokens can be approximately considered to occupy a union of several (identified as $K$) low-dimensional spaces, each with a dimension $p \ll d$. These spaces are characterized by orthonormal bases, represented as $\boldsymbol{U}_{[K]} = (\boldsymbol{U}_k)_{k=1}^K, \boldsymbol{U}_k \in \mathbb{R}^{d \times p}$. Within the framework of this local signal model, CRATE aims to optimize the *sparse rate raduction* objective:

$$\max_{f \in \mathcal{F}} \mathbb{E}_{\boldsymbol{Z}} \big[ \Delta R(\boldsymbol{Z} \mid \boldsymbol{U}_{[K]}) - \lambda \|\boldsymbol{Z}\|_0 \big] = \max_{f \in \mathcal{F}} \mathbb{E}_{\boldsymbol{Z}} \big[ R(\boldsymbol{Z}) - \lambda \|\boldsymbol{Z}\|_0 - R^c(\boldsymbol{Z}; \boldsymbol{U}_{[K]}) \big]. \quad (1)$$

where $\lambda$ is the sparsification regularizer and $\boldsymbol{Z} = f(\boldsymbol{X})$. The coding rate $R(\boldsymbol{Z})$ serves as a close estimate (following Ma et al. (2007)) for the average amount of bits necessary for encoding the tokens $\boldsymbol{z}_i$ to a precision level $\varepsilon$ using a Gaussian codebook. Additionally, $R^c(\boldsymbol{Z} \mid \boldsymbol{U}_{[K]})$ represents the theoretical maximum average amount of bits needed to encode the projection of the tokens onto each low dimensional subspace defined in the local signal model, specifically $\boldsymbol{U}_k^* \boldsymbol{z}_i$, to the same precision level $\varepsilon$ utilizing a Gaussian codebook, as outlined by Yu et al. (2023a). If the subspaces are adequately incoherent from each other, the solutions that minimize the object function, viz. Equation (1), in terms of $\boldsymbol{Z}$, are associated with subspace configurations that are both incoherent and aligned with the axes, as pointed out by Yu et al. (2020).

A network aimed at optimizing the sparse coding rate reduction objective through unrolled optimization gradually shifts the distribution of $\boldsymbol{X}$ towards the intended canonical forms, where each iteration of the unrolled optimization process acts as a layer.

$$f : \boldsymbol{X} \xrightarrow{f^{\mathrm{pre}}} \boldsymbol{Z}^1 \to \cdots \to \boldsymbol{Z}^\ell \xrightarrow{f^\ell} \boldsymbol{Z}^{\ell+1} \to \cdots \to \boldsymbol{Z}^{L+1} = \boldsymbol{Z} \xrightarrow{f^{\mathrm{head}}} \boldsymbol{Y}, \quad (2)$$

The iterative optimization framework incorporates multiple design choices, among which is a two-step alternating minimization approach grounded in robust theoretical principles (Yu et al., 2023a). This approach delineates two distinct blocks: the MSSA and the ISTA block, collectively defining a single CRATE layer:

$$\boldsymbol{Z}^{\ell+1/2} \doteq \boldsymbol{Z}^\ell + \mathrm{MSSA}(\boldsymbol{Z}^\ell \mid \boldsymbol{U}_{[K]}^\ell), \qquad f^\ell(\boldsymbol{Z}^\ell) = \boldsymbol{Z}^{\ell+1} \doteq \mathrm{ISTA}(\boldsymbol{Z}^{\ell+1/2} \mid \boldsymbol{D}^\ell). \quad (3)$$

**Compression operator: Multi-Head Subspace Self-Attention (MSSA).** Given local models $\boldsymbol{U}_{[K]}^\ell$, to form the incremental transformation $f^\ell$ optimizing Equation (1) at layer $\ell$, CRATE first compresses the token set $\boldsymbol{Z}^\ell$ against the subspaces by minimizing the coding rate $R^c(\cdot \mid \boldsymbol{U}_{[K]}^\ell)$. As Yu et al. (2023a) show, doing this minimization locally by performing a step of gradient descent on $R^c(\cdot \mid \boldsymbol{U}_{[K]}^\ell)$ leads to the so-called multi-head subspace self-attention (MSSA) operation, defined as

$$\mathrm{MSSA}(\boldsymbol{Z} \mid \boldsymbol{U}_{[K]}) \doteq \frac{p}{(N+1)\varepsilon^2} [\boldsymbol{U}_1, \ldots, \boldsymbol{U}_K] \begin{bmatrix} (\boldsymbol{U}_1^* \boldsymbol{Z}) \mathrm{softmax}\left((\boldsymbol{U}_1^* \boldsymbol{Z})^*(\boldsymbol{U}_1^* \boldsymbol{Z})\right) \\ \vdots \\ (\boldsymbol{U}_K^* \boldsymbol{Z}) \mathrm{softmax}\left((\boldsymbol{U}_K^* \boldsymbol{Z})^*(\boldsymbol{U}_K^* \boldsymbol{Z})\right) \end{bmatrix}, \quad (4)$$

In practice, the calculation of the intermediate representations $\boldsymbol{Z}^{\ell+1/2}$ with the output from the MSSA block is calculated in a weighted form:

$$\boldsymbol{Z}^{\ell+1/2} \approx \left(1 - \kappa \cdot \frac{p}{(N+1)\varepsilon^2}\right) \boldsymbol{Z}^\ell + \kappa \cdot \frac{p}{(N+1)\varepsilon^2} \cdot \mathrm{MSSA}(\boldsymbol{Z}^\ell \mid \boldsymbol{U}_{[K]}), \quad (5)$$

where $\kappa > 0$ is a learning rate hyperparameter. This block resembles to GPT's multi-head self-attention block, but the query, key, and value projection matrices within a single head are all identical in the MSSA block.

**Sparsification operator: Iterative Shrinkage-Thresholding Algorithm (ISTA).** The remaining term to optimize in Equation (1) is the difference of the global coding rate $R(\boldsymbol{Z})$ and the $\ell^0$ norm of the tokens, which together encourage the representations to be both sparse and non-collapsed. Yu

et al. (2023a) show that local minimization of this objective in a neighborhood of the intermediate representations $\boldsymbol{Z}^{\ell+1/2}$ is approximately achieved by a LASSO problem with respect to a sparsifying orthogonal dictionary $\boldsymbol{D}^{\ell} \in \mathbb{R}^{d \times h}$. Taking an iterative step towards solving this LASSO problem gives the iterative shrinkage-thresholding algorithm (ISTA) block (Wright and Ma, 2022; Yu et al., 2023a). The ReLU nonlinearity appearing in this block arises from an additional non-negativity constraint on the representations in the LASSO program, motivated by the goal of better separating distinct modes of variability in the token distribution:

$$\boldsymbol{Z}^{\ell+1} = f^{\ell}(\boldsymbol{Z}^{\ell}) = \mathrm{ReLU}(\boldsymbol{Z}^{\ell+1/2} + \eta \boldsymbol{D}^{\ell *}(\boldsymbol{Z}^{\ell+1/2} - \boldsymbol{D}^{\ell}\boldsymbol{Z}^{\ell+1/2}) - \eta\lambda\mathbf{1}) \doteq \mathtt{ISTA}(\boldsymbol{Z}^{\ell+1/2} \mid \boldsymbol{D}^{\ell}). \quad (6)$$

## 4 THE CRATE LANGUAGE MODEL

This section introduces the difference between our work and the original CRATE paper (Yu et al., 2023a), thus introducing what changes we made to the CRATE architecture. We first note that the task in this work is different: we apply the CRATE architecture to the *next-token prediction task* in the language domain. while the original CRATE paper applies the architecture to the image classification task in the vision domain. This difference leads to differences in the architecture design: **(i)** we apply a causal mask to the original CRATE model to avoid the model seeing tokens after the current token, and **(ii)** we change the embedding layer and heads of the original CRATE model.

Second, we're interested in interpreting the neurons within CRATE on the next-token prediction task and making direct comparisons to the GPT architecture. As neuron-level interpretation is commonly evaluated on the hidden states in the FFN block of the GPT model (Bills et al., 2023; Bricken et al., 2023), **(iii)** we increase the hidden dimension of the ISTA block of the original CRATE model to align with the hidden dimension of the FFN block of the GPT model. We thus call the new ISTA block the ISTA-overcomplete block.

Below we show specific definitions of the modifications we made. We illustrate the architecture in Figure 6, show implementation details in Appendix A, and discuss about the learning process in Appendix A.5.

**Embedding and Head.** In order to apply the CRATE architecture to the language domain, we define the pre-processing layer $f^{\mathrm{pre}}$ that transforms tokens into position-aware semantic embeddings, and define post-processing head $f^{\mathrm{head}}(\boldsymbol{Z})$ that maps the representations to output token distributions:

$$f^{\mathrm{pre}}(\boldsymbol{X}) = \boldsymbol{E}_{\mathrm{sem}}(\boldsymbol{X}) + \boldsymbol{E}_{\mathrm{pos}}, \qquad f^{\mathrm{head}}(\boldsymbol{Z}) = \boldsymbol{W}^{\mathrm{head}}\boldsymbol{Z}, \quad (7)$$

where $\boldsymbol{E}_{\mathrm{sem}}$ is a semantic embedding matrix that maps input tokens $\boldsymbol{x}_i$ to embedding vectors in $\mathbb{R}^d$, $\boldsymbol{E}_{\mathrm{pos}} \in \mathbb{R}^{d \times N}$ is a positional embedding matrix, and $\boldsymbol{W}^{\mathrm{head}} \in \mathbb{R}^{V \times d}$ maps the (contextualized) token representations $\boldsymbol{Z}^{L+1}$ to the distribution of the next token. All parameters mentioned are learnable.

**MSSA Block.** To align with the next word prediction task used in GPT (Radford et al., 2019), we replace the attention matrix in MSSA (Equation (4)) with a *causally masked* self-attention, defined as

$$\mathrm{softmax}((\boldsymbol{U}_k^*\boldsymbol{Z})^*(\boldsymbol{U}_k^*\boldsymbol{Z})) \to \mathrm{softmax}\left(\mathtt{CausalMask}((\boldsymbol{U}_k^*\boldsymbol{Z})^*(\boldsymbol{U}_k^*\boldsymbol{Z}))\right),$$

$$\text{where } \mathtt{CausalMask}(\boldsymbol{M})_{ij} = \begin{cases} \boldsymbol{M}_{ij}, & i \leq j \\ -\infty, & i > j. \end{cases} \quad (8)$$

**ISTA Block.** To investigate the neuron interpretability of the activation matrix $\boldsymbol{A} \in \mathbb{R}^{h \times N}$, we design an *overcomplete* version of the ISTA block (Equation (6)) with $\boldsymbol{D}^{\ell} \in \mathbb{R}^{d \times h}$ where $h = nd$, and $n = 4$ to keep a fair comparison to GPT (also same as standard transformer architecture proposed in Vaswani et al. (2017)):

$$\boldsymbol{A}_t \doteq \mathtt{ISTA}(\boldsymbol{Z}^{\ell+1/2} \mid \boldsymbol{D}^{\ell}),$$

$$\boldsymbol{A}_k = \mathrm{ReLU}(\boldsymbol{A}_{k-1} - \eta(\boldsymbol{D}^{\ell})^*(\boldsymbol{D}^{\ell}\boldsymbol{A}_{k-1} - \boldsymbol{Z}) - \eta \cdot \lambda \cdot \mathbf{1}), \quad \boldsymbol{A}_0 = \mathbf{0}, \ k \in [t], \quad (9)$$

$$\boldsymbol{Z}^{\ell+1} = \boldsymbol{D}^{\ell}\boldsymbol{A}_t$$

Here, $\eta > 0$ is the step size, $\lambda > 0$ is the sparsification regularizer, and $t$ is the number of ISTA iterations. In practice, we set $t = 2$ to keep computation efficient. The ISTA block resembles the MLP block in the GPT model, but with a relocated skip connection.

## 5 EMPIRICAL EXPERIMENTS

We examine the next token prediction performance and neuron interpretability of CRATE in this section. We detail the architecture, size, pre-training recipe (Section 5.1), performance (Section 5.2), and neuron-level interpretability (Section 5.3) of CRATE compared to the standard transformer architecture. In this section, we denote $K$ as the number of attention heads, $d$ as the dimension of the residual stream in the model, and $h$ as the hidden (inner) dimension of the ISTA/MLP module.

### 5.1 SETUP

**Model architecture and size.** The CRATE model is designed with various sizes $L \in \{1, 2, 3, 6, 12\}$, where each size matches the GPT configurations for direct comparisons, as shown in Section 5.1. Configurations for $L \in \{1, 2, 3\}$ adhere to GPT models as per Bricken et al. (2023), while $L \in \{6, 12\}$ follow configurations from Sanh et al. (2019) and Radford et al. (2019), respectively. Notably, CRATE maintains approximately $2/3$ the size of GPT at scale. Both models utilize the Byte-level BPE tokenizer with a 50,257 vocabulary size, following Radford et al. (2019). We explain the difference between CRATE and GPT in parameter size to GPT in Appendix A.1.

| Model Config | $d$ | $K$ | $L$ | $h$ | CRATE | GPT |
|---|---|---|---|---|---|---|
| **1L** | 128 | 4 | 1 | 512 | 6.54M | 6.64M |
| **2L** | 128 | 4 | 2 | 512 | 6.64M | 6.83M |
| **3L** | 128 | 4 | 3 | 512 | 6.74M | 7.03M |
| **S**(mall) | 768 | 12 | 6 | 3,072 | 55.9M | 81.1M |
| **B**(ase) | 768 | 12 | 12 | 3,072 | 81.2M | 123.6M |

Table 1: Model configuration of CRATE and model size comparison to GPT.

**Datasets and optimization.** We pre-train both models using the next token prediction task on the uncopyrighted *Pile* dataset (Gao et al., 2020) using the Adam optimizer (Kingma and Ba, 2015). Following Bricken et al. (2023), we pre-train both CRATE and GPT of smaller sizes ($L \in \{1, 2, 3\}$) on 100 billion tokens with a context window of 1,024 tokens. Following the pre-training setup in Karpathy (2022) and scaling law in Touvron et al. (2023), we pre-train using 100 billion tokens for the Small models, and 160 billion tokens for the Base models.[2] It takes 4 days to pre-train CRATE-Base on 160 billion tokens with 32 A5000 GPUs.

### 5.2 PERFORMANCE

This section demonstrates that CRATE, despite not outperforming GPT-2, still generates reasonable predictions, as evidenced through quantitative and qualitative comparisons.

We observe that both training and validation loss curve of CRATE-Base on the Pile dataset *converges* well, as presented in Figure 2 (*left*). Although the convergence is slower than GPT, the loss curve of CRATE keeps decreasing after training on 160 billion tokens, while GPT already tends to converge.

We also demonstrate the zero-shot validation loss curve of CRATE evaluated on OpenWebText as well as other datasets (Radford et al., 2019) in Figure 2 (*right*). Results show that CRATE effectively learns transferable representations across a number of datasets, and achieves comparable performance to GPT after full training on the 160 billion tokens. We also demonstrate the *scalability* of the CRATE architecture by comparing the validation loss of CRATE and GPT with respect to the model size in Figure 3 (*left*). Results show that the performance of CRATE is close to GPT across all model sizes. However, we do recognize that forcing sparsification in a model potentially leads to a higher compute cost on the next-token-prediction objective, which aligns with observations in Bricken et al. (2023) that enabling monosemanticity might hurt model performance.

Qualitative examples from CRATE and GPT are demonstrated in Figure 3 (*right*). We conclude that CRATE can make reasonable predictions, encouraging us to further look into its neuron-level interpretability.

---

[2] Practicaly, we train with a batch size of 768 for 125,000 iterations for $L \in \{1, 2, 3, 6\}$, and a batch size of 256 for 600,000 iterations for $L = 12$.

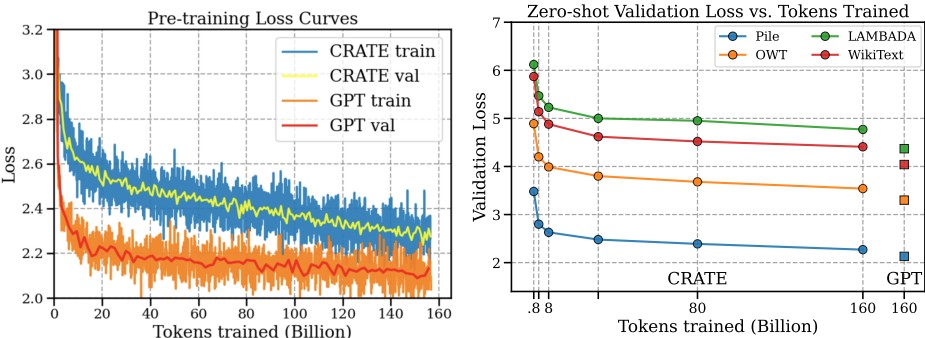

Figure 2: *Left:* loss curve when pre-training CRATE-Base and GPT-Base on the Pile dataset. *Right:* zero-shot validation loss of CRATE evaluated on a variety of datasets (Pile, LAMBADA, OpenWebText and WikiText).

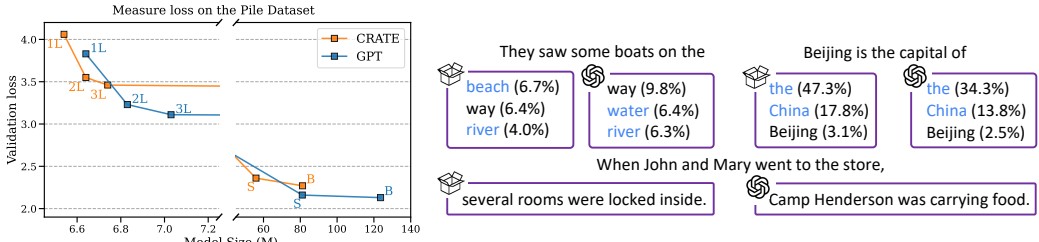

Figure 3: *Left:* Validation loss of CRATE compared to GPT on the Pile dataset, with respect to the model size. *Right:* Qualitative examples of predictions made by our models and the official models. The tokens in blue are considered good. We compare CRATE-Base to GPT2-Base on the next word prediction task.

## 5.3 INTERPRETABILITY

In order to quantitatively evaluate the interpretability of the neuron activations, we adopt the large language model-based approach introduced in Bills et al. (2023) and Bricken et al. (2023). We demonstrate the algorithm for scoring interpretations in Algorithm 1. We retrieve the sparse code $A_t$ (activations after the ReLU unlinearlity in the ISTA block) of CRATE for interpretation, and compare with activations from the MLP block of GPT.

---

**Algorithm 1** Interpretability Evaluation Algorithm (Bills et al., 2023)

---

1: **Inputs:** Input token set $S$ (in text form) and its activation matrix $A \in \mathbb{R}^{h \times T \times B}$ at $\ell$-th layer, where $T$ is the length of a single text excerpt, and $B$ is the number of text excerpts in the corpus.
2: **Models:** Explanation model $\mathcal{F}_1$ , simulation model $\mathcal{F}_2$.
3: **for** $i \in [d]$ **do**
4:   $S' \sim S, A' \in \mathbb{R}^{h \times T \times b} \sim A$ : *Retrieve* $b$ text excerpts of $T$ tokens, together with the corresponding activation matrices.
5:   $k_i = \mathcal{F}_1(S', A'_{i,*})$ : *Explain* common patterns retrieved activations of $i$-th neuron.
6:   $\tilde{A}'_{i,*} = \mathcal{F}_2(k_i, S')$ : Use the explanation to *simulate* scores given only the tokens, not including true activations.
7:   $\rho_i = \rho(A'_{i,*}, \tilde{A}'_{i,*})$ : Calculate *correlation* between the accurate and simulated activations.
8: **end for**
9: **Output:** Averaged interpretation score over all neurons $s = \mathbb{E}_{i \in [d]}(\rho_i)$.

---

In practice, we adopt three evaluation metrics: two from OpenAI (*top-and-random* and *random-only*) (Bills et al., 2023) and one from Anthropic (Bricken et al., 2023). We adopt the official implementation from Wu et al. (2023), where details on the implementation are elaborated in Appendix B. Note that the Anthropic metric has much shorter text excerpts than the OpenAI metrics, so it is biased to sparse activations. For all evaluations, we discard the last layer of CRATE and

| | Mean (↑, darker green means more interpretable) | | | | | | Variance (↓, darker red means less steady) | | | | | |
|---|---|---|---|---|---|---|---|---|---|---|---|---|
| | Top-and-Random | | Random-only | | Anthropic | | Top-and-Random | | Random-only | | Anthropic | |
| | CRATE | GPT | CRATE | GPT | CRATE | GPT | CRATE | GPT | CRATE | GPT | CRATE | GPT |
| 1L | 3.9 | 8.8 | 4.8 | 8.9 | 10.1 | 14.2 | 0.0 | 0.0 | 0.0 | 0.0 | 0.0 | 0.0 |
| 2L | 8.05 | 4.2 | 6.95 | 1.95 | 11.35 | 10.2 | 0.06 | 0.01 | 1.1 | 0.12 | 0.0 | 0.25 |
| 3L | 9.1 | 3.57 | 8.43 | 1.37 | 11.23 | 9.2 | 0.26 | 7.51 | 1.2 | 1.93 | 1.14 | 19.21 |
| 6L | 7.96 | 5.4 | 6.36 | 3.14 | 10.4 | 8.52 | 2.29 | 20.85 | 1.87 | 18.39 | 2.01 | 32.56 |
| 12L | 6.8 | 6.34 | 5.12 | 2.67 | 8.88 | 8.65 | 7.09 | 11.35 | 2.83 | 7.48 | 18.3 | 24.65 |

Table 2: Mean and variance of the average interpretability across layers for different model sizes.

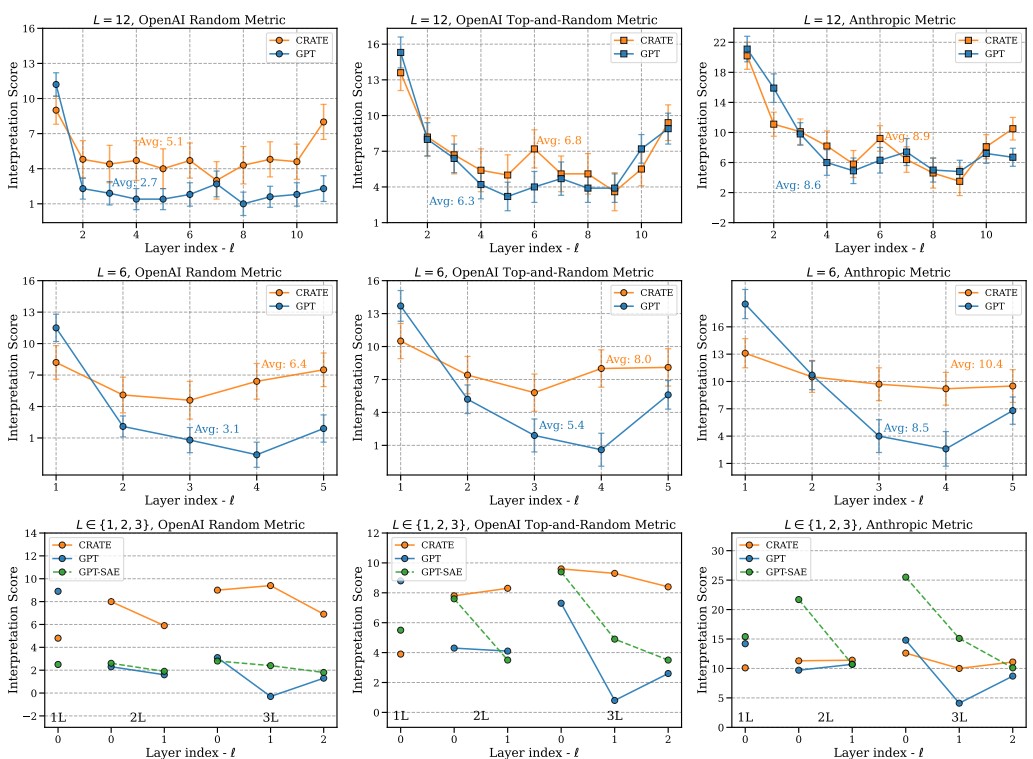

Figure 4: Interpretation scores evaluated using the OpenAI Random-only metric, Top-and-Random metric, and Anthropic metric, respectively. *Top:* interpretation scores of CRATE and GPT for $L = 12$. *Middle:* interpretation scores of CRATE and GPT for $L = 6$. *Bottom:* interpretation scores of CRATE, GPT, and GPT with sparse auto-encoder for $L \in \{1, 2, 3\}$. Variance bars are normalized to $1/10$ of its original size.

GPT, according to the empirical observation that the last layer of CRATE is biased to the pre-training task (Yu et al., 2023b).

**CRATE achieves markedly improved and more steady neuron-level interpretability across layers compared to GPT-2, applicable across a wide range of model sizes.** We show evaluation results of the interpretability of CRATE and GPT averaged across layers in Table 2 (*left*). We observe that the interpretability of CRATE comprehensively outperforms GPT on all metrics for $L \in \{2, 3, 6, 12\}$. When averaging the mean interpretability across all metrics, CRATE outperforms GPT up to strikingly 45.1% when $L = 6$, and up to 16.3% when $L = 12$. We also present the layer-wise interpretation scores in Figure 4, which shows that CRATE has higher interpretability than GPT on almost all layers using the OpenAI metrics, and is slightly better than GPT using the Anthropic metric. For detailed distributions of the layer-wise scores of CRATE-Base compared to GPT-Base on different metrics, refer to Appendix D.

The variances of the average interpretation scores of CRATE and GPT across layers are shown in Table 2 (*right*). From the results we draw a solid conclusion that the interpretability of CRATE is much more steady than GPT across all model sizes. Figure 4 further demonstrates a clear pattern that, for all model sizes, CRATE maintains a higher interpretability than GPT among almost all layers.

**The built-in sparse coding approach introduces consistent and specific neuron-level behaviors.** The strong interpretability of CRATE on the OpenAI top-and-random metric and the Anthropic metric, as shown in Figure 4, indicates its consistent behavior on relevant tokens. These two methods contain a large portion of top-activated text excerpts, so they are valid for measuring whether the activations are consistent with the summarized explanation (Bills et al., 2023; Bricken et al., 2023). Additionally, the larger interpretability gap of CRATE and GPT on the OpenAI random-only metric versus the top-and-random metric highlights the specificity of CRATE in avoiding firing on irrelevant tokens. The random-only metric usually includes highly irrelevant text excerpts, so it effectively measures the capability of the language model to avoid activating on semantically irrelevant tokens (Bills et al., 2023).

Qualitatively, we refer back to the qualitative examples shown in Figure 1. We list three neurons from CRATE (row 1) and GPT (row 2), respectively. For each neuron, we show two top-activated text excerpts and one random excerpt. Results show that CRATE is able to consistently activate on semantically similar tokens within the most relevant text excerpts, and does not activate on random tokens that are semantically distinguished from the top tokens. This promotes a more precise explanation given by the explanation model (Mistral in the figure). On the other hand, GPT is much worse at distinguishing tokens from different contexts, because it also has high activations on random text excerpts where the semantic meanings deviate from the top activations a lot. As a side note, we also analyze the activation sparsity of CRATE and GPT in Appendix C.

**Comparing CRATE to GPT with post-hoc sparse auto-encoders.** We follow Bricken et al. (2023) and train SAEs for models with layers $L \in \{1, 2, 3\}$, using output activations from GPT on the Pile dataset's training split, leading to the GPT-SAE model. Details on the SAEs' architecture and training are in Appendix E.

The interpretability scores of GPT-SAE compared to CRATE and GPT, as depicted in Figure 4, reveal that under the long-context OpenAI metrics, GPT-SAE matches GPT but falls short of CRATE. This is attributed to its neuron activations becoming nearly 99% sparse after sparse auto-encoding, diminishing interpretability in long contexts. Conversely, under the Anthropic metric, GPT-SAE surpasses both GPT and CRATE in interpretability, corroborating findings in Bricken et al. (2023) that post-hoc approaches enhance short-context interpretability, often a sign of mono-semanticity. However, the interpretability of GPT-SAE on the Anthropic metric decreases significantly when $\ell$ increases, while CRATE remains steady. Further qualitative comparisons are can be found in Appendix F.

Besides its good performance on the Anthropic metric, the post-hoc dictionary learning approach requires considerable *manual effort*. To get a taste, training a sparse auto-encoder for a single GPT layer takes 4 hours when $h = 512$ and a day when $h = 3072$ on an A100 GPU.

|  | OpenAI TaR | Anthropic |
|---|---|---|
| CRATE-SAE - CRATE | **-10.2** | **+34.8** |
| GPT-SAE - GPT | +6.5 | +38.1 |

Table 3: **Interpretability improvement of CRATE and GPT after applying SAE.** Results are obtained by subtracting the interpretation scores of the language model and the SAE model trained on that language model. Results consistently show that the interpretability improvement of CRATE-SAE over CRATE is smaller than GPT-SAE over GPT, indicating more optimal representations of CRATE over GPT.

**Does CRATE have more optimal representation than GPT in terms of interpretability?** Alternatively, we train SAE models upon the CRATE model, and compare the interpretability improvement of CRATE-SAE over CRATE to the interpretability improvement of GPT-SAE over GPT. As shown in Table 3, the improvement of interpretability of CRATE-SAE over CRATE is smaller than GPT-SAE over GPT under both OpenAI and Anthropic metrics. This suggests that CRATE has more optimal representations than GPT in terms of interpretability. Experimental details can be found in Appendix E.

## 5.4 DISENTANGLING INTERPRETABILITY FROM OTHER FACTORS

**Is the improved interpretability due to performance gap?** We first investigate whether the interpretability improvement is due to worse performance by comparing the interpretability of two different checkpoints of CRATE (intermediate checkpoint and full checkpoint). Results in Table 4

| Checkpoint | Loss | 0 | 1 | 2 | 3 | 4 | 5 | 6 | 7 | 8 | 9 | 10 | 11 | Avg |
|---|---|---|---|---|---|---|---|---|---|---|---|---|---|---|
| 79B Tokens | 2.38 | 13.9 | 8.3 | 6.5 | 6.0 | 4.3 | 6.7 | 5.1 | 4.6 | 3.7 | 5.2 | 8.0 | 2.8 | 6.3 |
| 158B Tokens | 2.29 | 13.6 | 8.2 | 6.7 | 5.4 | 5.0 | 7.2 | 5.1 | 5.1 | 3.6 | 5.5 | 9.4 | 5.3 | 6.7 |

Table 4: **Validation loss and interpretability of the CRATE model at different checkpoints.** The interpretation scores are under the OpenAI TaR metric.

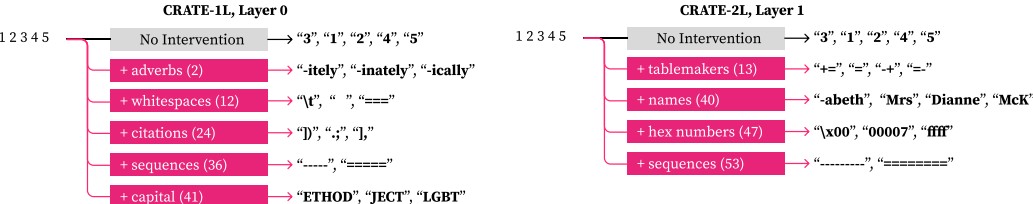

Figure 5: **Qualitative examples on logit effects of manually activating feature (i) in CRATE.** Text shown on the right side are the most positive changes in token prediction probability. The logit effects align with feature interpretations.

show that the interpretability of the intermediate checkpoint is lower than the full checkpoint, which suggests that "sacrificing" performance does *not* necessarily introduce better interpretation scores.

Another piece of evidence is that CRATE-2L has higher interpretability scores than CRATE-1L. As shown in Figure 3 (*left*), CRATE-2L has much better performance than CRATE-1L on the next-token prediction task. On the other hand, as shown in Table 2, the interpretability of CRATE-2L is also much higher than CRATE-1L. Thus, a lower performance does not necessarily introduce higher interpretability.

**Is the interpretability gap due to number of parameters?** We observe that CRATE-Base (81.2M) has a similar number of parameters as GPT-Small (81.1M). However, results in Table 2 indicate that the interpretability of CRATE-Base is higher than GPT-Small on all metrics, and their layer-wise interpretation scores are also different in Figure 4. This evidence suggests that two models with similar number of parameters does not necessarily have similar interpretability.

Another piece of evidence is that the interpretability of CRATE/GPT-1L all the way up to CRATE/GPT-12L does not have a consistent trend of increasing/decreasing interpretability, but their number of parameters both monotonously increases. This indicates that a model with larger number of parameters does not necessarily has better/worse interpretability.

**Steering the CRATE model.** Following Bricken et al. (2023), we manually activate some neurons and observe the *logit effects* (changes of the token probability of the language model head). Some qualitative examples are shown in Figure 5. Compared to the lossy steering of the SAE models, CRATE are steered without loss. Discussions on the lossy steering process can be found in Appendix E.6.

# 6 CONCLUSION, LIMITATION, AND FUTURE WORK

In this paper, we demonstrated that replacing the standard transformer architecture with the white-box model CRATE as a foundational architecture significantly improves the interpretability. Our empirical findings on the capability of CRATE to be consistent and distinctive on the neuron-level activations underscore the importance of the white-box design in developing better language foundation models, fostering optimism that the introduction of built-in sparse coding approaches will catalyze further advancements in neuron-level interpretations.

Despite these findings, we acknowledge that the performance of CRATE is not as good as GPT on the next-token prediction task, which is potentially due to the introduction of the ISTA operator that introduces sparsity. This aligns with previous work suggesting that the performance might drop when explicitly introducing sparsity (Bricken et al., 2023). Future work should investigate towards a better trade-off between performance and interpretability of language models with built-in sparsity. It would also be meaningful to research on more qualitative mechanisms in the white-box language model and how to use these mechanisms for downstream edits.

## REPRODUCIBILITY STATEMENT

To facilitate reproducibility of our work, we will open-source the model checkpoints and training infrastructure. We have included the model architecture in Section 4, pre-training recipe (including dataset and hyper-parameters) in Section 5.1, interpretability evaluation in Section 5.3, and SAE training setup in Appendix E.2.

## ETHICS STATEMENT

By improving the interpretability of language models, our work promotes a deeper understanding of their mechanisms, aiding in the identification and mitigation of potential risks, thereby supporting transparency and responsible AI development. On the language model side, this research pretrains a GPT-2-sized language model on publicly available data, with no intentional inclusion of harmful content. The model's moderate size and data scope reduce the likelihood of generating harmful or out-of-distribution outputs. However, risks associated with intentionally training models on harmful datasets, which can lead to biased or unsafe generations, must be considered.

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

## A DETAILS ON THE CRATE ARCHITECTURE

### A.1 PARAMETER SIZE OF CRATE AND GPT

CRATE is smaller than GPT because of the architecture difference. The vanilla GPT architecture has two main parameterized blocks: Attention block and MLP block.

**Parameter size of the MSSA Block.** In CRATE, the MSSA block resembles the Attention block, but instead of K, Q, V matrices, we only have one matrix. Therefore, compared to standard transformers, CRATE uses $1/3$ of the parameters for the multi-head attention part.

**Parameter size of the ISTA block.** The MLP block in vanilla GPT has one parametric matrix that transforms the input representations to the inner space (usually 4x larger), and another parametric matrix that transforms the inner representations back to the output space (as large as the input space). In CRATE, the MLP block is replaced by the ISTA-overcomplete block, which transforms the input representation to the overcomplete basis (4x larger) and transforms back with the same parametric matrix. Therefore, compared to standard transformers, CRATE uses $1/2$ of the parameters for the MLP part.

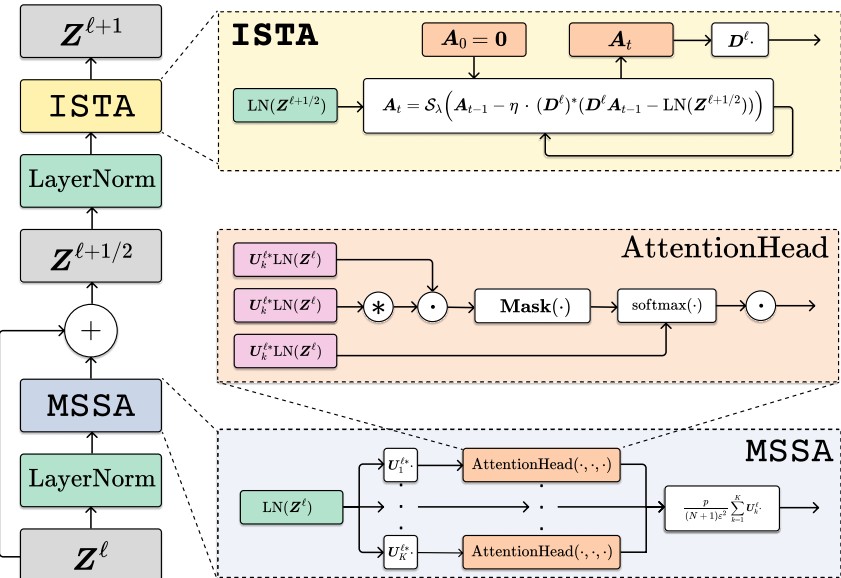

Figure 6: Block architecture for the CRATE language model, where $S_\lambda(x) = \text{ReLU}(x - \eta \cdot \lambda \cdot 1)$. Differences from the original architecture mentioned in Yu et al. (2023a) are marked **bold**: we (1) add a causal mask **Mask**$(\cdot)$ and (2) over-parameterize the **ISTA** block.

## A.2 Overall Architecture

The overall architecture is visualized in Figure 6.

## A.3 Causal MSSA Block

This process can be implemented by PyTorch-like code shown in Algorithm 2.

---

**Algorithm 2** PyTorch-Like Code for Causal MSSA Forward Pass

---

```
class CausalMSSA(nn.Module):
    def __init__(self, config):
        super().__init__()
        assert config.n_embd % config.n_head == 0
        self.c_attn = nn.Linear(config.n_embd, config.n_embd, bias=False)
        self.c_proj = nn.Linear(config.n_embd, config.n_embd, bias=True)
        self.attn_dropout = nn.Dropout(config.dropout)
        self.resid_dropout = nn.Dropout(config.dropout)
        self.n_head = config.n_head
        self.n_embd = config.n_embd
        self.dropout = config.dropout
        self.register_buffer("bias", torch.tril(torch.ones(config.block_size, config.
        block_size)).view(1, 1, config.block_size, config.block_size)) # causal mask

    def forward(self, x, enhanced_feature_id=None):
        B, T, C = x.size()
        qkv = qkv.view(B, T, self.n_head, C // self.n_head).transpose(1, 2)
        att = (qkv @ qkv.transpose(-2, -1)) * (1.0 / math.sqrt(qkv.size(-1)))
        att = att.masked_fill(self.bias[:,:,:T,:T] == 0, float('-inf'))
        att = F.softmax(att, dim=-1)
        att = self.attn_dropout(att)
        y = att @ qkv
        y = y.transpose(1, 2).contiguous().view(B, T, C)
        y = self.resid_dropout(self.c_proj(y))
        return y
```

---

### A.4 OVER-COMPLETE ISTA BLOCK

To give a better idea of how Equation (9) works, we expand the two-iteration process ($t = 2$). Given $\boldsymbol{D}^\ell \in \mathbb{R}^{d \times h}$, we expand the first ISTA step to

$$
\begin{aligned}
\boldsymbol{A}_0 &= \boldsymbol{0}, \\
\boldsymbol{A}_1 &= \mathcal{S}_\lambda \left( \boldsymbol{A}_0 - \eta \cdot (\boldsymbol{D}^\ell)^* (\boldsymbol{D}^\ell \boldsymbol{A}_0 - \mathrm{LN}(\boldsymbol{Z}^{\ell+1/2})) \right) \\
&= \mathrm{ReLU} \left( \eta \cdot (\boldsymbol{D}^\ell)^* \mathrm{LN}(\boldsymbol{Z}^{\ell+1/2}) - \eta\lambda \right).
\end{aligned}
\tag{10}
$$

The second ISTA step continues the process from the initialized sparse code $\boldsymbol{A}_1$:

$$
\begin{aligned}
\boldsymbol{A}_2 &= \mathcal{S}_\lambda \left( \boldsymbol{A}_1 - \eta \cdot (\boldsymbol{D}^\ell)^* (\boldsymbol{D}^\ell \boldsymbol{A}_1 - \mathrm{LN}(\boldsymbol{Z}^{\ell+1/2})) \right) \\
&= \mathrm{ReLU} \left( \boldsymbol{A}_1 - \eta \cdot (\boldsymbol{D}^\ell)^* (\boldsymbol{D}^\ell \boldsymbol{A}_1 - \mathrm{LN}(\boldsymbol{Z}^{\ell+1/2})) - \eta\lambda \right),
\end{aligned}
\tag{11}
$$

which can be decomposed to:

$$
\begin{aligned}
\boldsymbol{G}_1 &= (\boldsymbol{D}^\ell)^* \boldsymbol{D}^\ell \boldsymbol{A}_1 \\
\boldsymbol{G}_2 &= (\boldsymbol{D}^\ell)^* \cdot \mathrm{LN}(\boldsymbol{Z}^{\ell+1/2}) \\
\boldsymbol{G} &= \eta \cdot (\boldsymbol{G}_2 - \boldsymbol{G}_1) - \eta \cdot \lambda \\
\boldsymbol{A}_2 &= \mathrm{ReLU}(\boldsymbol{A}_1 + \boldsymbol{G})
\end{aligned}
\tag{12}
$$

where $\boldsymbol{A}_2$ is the output sparse code. At last, we convert the output sparse code from the coding rate space back to the original representation space:

$$
\boldsymbol{Z}^{\ell+1} = \boldsymbol{D}^\ell \boldsymbol{A}_2
\tag{13}
$$

This process can be implemented by PyTorch-like code shown in Algorithm 3.

---

**Algorithm 3** PyTorch-Like Code for Over-complete ISTA Forward Pass

---

```
1  class ISTA(nn.Module):
2      def __init__(self, config):
3          super().__init__()
4          self.weight = nn.Parameter(torch.Tensor(4 * config.n_embd, config.n_embd)) # h*d
5          with torch.no_grad():
6              init.kaiming_uniform_(self.weight)
7          self.step_size = 0.1
8          self.lambd = 0.1
9
10     def forward(self, x, enhanced_feature_id=None):
11         z_init = F.relu(self.step_size * F.linear(x, self.weight, bias=None) - self.
       step_size * self.lambd) # A1
12         x1 = F.linear(z_init, self.weight.t(), bias=None)
13         grad_1 = F.linear(x1, self.weight, bias=None)
14         grad_2 = F.linear(x, self.weight, bias=None)
15         grad_update = self.step_size * (grad_2 - grad_1) - self.step_size * self.lambd
16         output_sparse_code = F.relu(z_init + grad_update) # A2
17         output = F.linear(output_sparse_code, self.weight.t(), bias=None)
18         return output
```

---

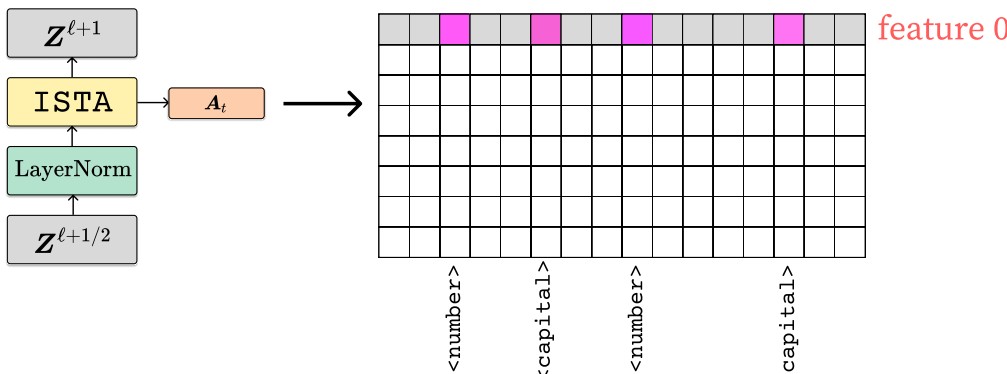

Figure 7: CRATE iteratively compresses (MSSA block) and sparsifies (ISTA block) the token representations (*colored points*) across its layers from 1 to $L$, transforming them into parsimonious representations aligned on axes (*colored lines*) with distinct semantic meanings.

## A.5 DETAILS OF THE LEARNING PROCESS

The desired optimization process is illustrated in Figure 7. The process starts with random token representations ($\mathbf{Z}^1$). Through successive layers, the representations ($\mathbf{Z}^\ell$) are *compressed* to align with the axis via the MSSA block, forming $\mathbf{Z}^{\ell+1/2}$ that are semantically more consistent among relevant tokens. This is then refined by *sparse coding* (the ISTA block) to produce the representations $\mathbf{Z}^{\ell+1}$ that align on incoherent axes, leading to semantically more specified token representations. Repeated across layers, this culminates in distinct token representations $Z^{L+1}$ aligned on unique semantic axes. More detailed explanation of this optimization process can be found in Appendix A.5.

We elaborate the learning process of CRATE in this section, with a close reference to Figure 7.

**In Figure 7, what is the space the points are drawn in?** The space is the representation space (of layer $\ell$). Because the model is pretrained with next-token prediction in the language domain, the space is specifically a semantic space. Thus, each point (token) has a semantic representation in this high-dimensional space.

Figure 8: Illustration of the concepts of the activation matrix and poly-semanticity.

**How do the layouts of the points suggest mono or poly-semanticity?** First, each axis (red/yellow) in the figure represents a neuron/feature visualized in the semantic space. We visualized the activation matrix $\mathbf{A}_t$ in Figure 8. For example, when $L \in \{1, 2, 3\}$, the model dimension is 128, which means that the overcomplete basis of ISTA will have a dimension of 512, introducing 512 features. Now if we input a sequence of 256 tokens, the activation matrix will have a shape of $[512, 256]$.

*Poly-semanticity* means that token representations in the semantic space are clustered as a broader set of semantic meanings - that is, each neuron has a broader set of semantic meanings. For example, in the gray box on the top left of Figure 7, both yellow- and red-backgrounded tokens represent either a number or a capitalized token. This corresponds to multiple high activations in the feature, where the tokens that activated this feature can either be a number or a capitalized token, which is shown in Figure 8 (where pink squares represent high activations).

In the *compression* phase, the token representations are pushed towards the semantic axes, so that the tokens will activate on fewer features but will gain higher activations on these features - which is essentially an activation condensing process.

In the *sparsification* phase, the neurons (axes) are made further from each other, meaning that the features have less semantic overlap with each other. In this case, the results will become the gray box on the top right side of Figure 7, indicating that each neuron has distinct semantic meanings, like "numbers" or "capitalized tokens".

Note that this is a minimal example. In practice, tokens appear in context.

## B  DETAILS ON INTERPRETABILITY EVALUATIONS

This section details the implementation details of the interpretability evaluations.

In practice, we adopt three evaluation metrics: two from OpenAI (Bills et al., 2023) and one from Anthropic (Bricken et al., 2023). As the Anthropic metric is a closed-source follow-up of OpenAI, we start from the official implementation provided by OpenAI (Wu et al., 2023) for both metrics.

For each layer, we use randomly sampled $8,000$ text excerpts of $1,024$ tokens each, which sums up to 8M tokens in total, from the test split of the uncopyrighted Pile dataset, to evaluate the interpretability scores.

### B.1  PARAMETERS OF EVALUATION METRICS

Comprehensive parameter settings are shown in Table 5. For the OpenAI metrics, each input text excerpt contains 64 tokens. For the *OpenAI top-and-random* metric, we use 5 top activated excerpts for explanation, and a mixture of 5 top activated and 5 randomly activated excerpts for simulation. For the *OpenAI random-only* metric, we only use 5 randomly activated excerpts for simulation.

For the *Anthropic* metric, each text excerpt contains only 8 tokens. For the explanation model, we input 15 top activated excerpts, 5 randomly activated excerpts, and 22 excerpts from different activation quantiles. To elaborate, we evenly divide the activation range into 11 quantiles, where we pick 2 excerpts from each of them. For the simulation model, we input 10 top activated excerpts, 5 randomly activated excerpts, 22 quantiled excerpts, and 10 top activated out-of-context (OOC) excerpts. Our implementation of the OOC excerpts is to cut the input text excerpt into length of only 3 tokens.

Table 5: Evaluation parameter settings of the OpenAI and Anthropic approach.

|  |  | #Token | Explanation | | | Simulation | | | |
| --- | --- | --- | --- | --- | --- | --- | --- | --- | --- |
|  |  |  | #Top | #Rand | #Qua | #Top | #Rand | #Qua | #OOC |
| **OpenAI** | TaR | 64 | 5 |  |  | 5 | 5 |  |  |
|  | Rand | 64 | 5 |  |  |  | 5 |  |  |
| **Anthropic** |  | 8 | 15 | 5 | $2 \cdot 11$ | 10 | 5 | $2 \cdot 11$ | 10 |

### B.2  DISCUSSION ON FOCUS OF DIFFERENT MEASURES

The OpenAI random-only metric is the easiest to interpret. As noted by Bills et al. (2023), the random-only metric considers an explanation's ability to capture the neuron's representation of features in the pre-training distribution, because the simulated tokens are uniformly randomly sampled from the validation set of the pre-train dataset. However, the random-only scoring with small sample size risks failing to capture behavior, due to lacking both tokens with high simulated activations and tokens with high real activations. Top-and-random scoring addresses the latter, but causes us to penalize falsely low simulations more than falsely high simulations, and thus tends to accept overly broad explanations.

The Anthropic metric, on the other hand, puts more focus on the mono-semanticity of the activations, as noted by Bricken et al. (2023). For sparse features, which don't fire on most random samples, evaluating across a wide range of activations effectively tests the model's ability to distinguish a

feature's large activations from zero, and the short text excerpts make it easier for the simulation model to identify the sparse activations.

## B.3 MORE ACCESSIBLE EVALUATION

To reduce compute cost, we use `Mistral-7B-instruct` as the explanation model, and `LLaMA-2-7B` as the simulation model. We empirically prove that these replacements does not affect the conclusions of apple-to-apple comparison between CRATE and GPT below.

**Explanation model.** In the official implementation (Wu et al., 2023), the explanation model is `gpt-4`. According to ablations described in Bills et al. (2023), it also makes sense to use the sligtly cheaper model `gpt-3.5-instruct`. Due to the high compute cost, we use the open-source model `mistral-7b-instruct` instead. We demonstrate the performance of `gpt-3.5-turbo` and `mistral-7b-instruct` using the OpenAI random-only and top-and-random metrics in Table 6. Results show that the change of model doesn't significantly change the scores, and doesn't affect conclusions at all.

Table 6: Interpretability measure of GPT, GPT-SAE and CRATE-GPT on the Pile dataset based on the OpenAI metrics. Explanation model: `Mistral-7B-instruct`/`GPT-3.5-turbo`. Simulation model: `LLaMA-2-7B`.

| mistral-7b-instruct | | | $\rho$ (**Random-only**) $(\%, \uparrow)$ | | | $\rho$ (**Top-and-Random**) $(\%, \uparrow)$ | | |
|---|---|---|---|---|---|---|---|---|
| Model | Size | Loss | Layer 1 | Layer 2 | Layer 3 | Layer 1 | Layer 2 | Layer 3 |
| CRATE-1L | 6.54M | 4.06 | 4.8 | - | - | 3.9 | - | - |
| CRATE-2L | 6.64M | 3.55 | 8.0 | 5.8 | - | 7.8 | 8.3 | - |
| CRATE-3L | 6.74M | 3.46 | 9.0 | 9.4 | 6.9 | 9.6 | 9.3 | 8.4 |
| GPT-1L | 6.64M | 3.83 | 8.9 | - | - | 8.8 | - | - |
| GPT-2L | 6.83M | 3.23 | 2.3 | 1.6 | - | 4.3 | 4.1 | - |
| GPT-3L | 7.03M | 3.11 | 3.1 | −0.3 | 1.3 | 7.3 | 0.8 | 2.6 |
| GPT-1L (16x SAE) | | | 2.9 | - | - | 5.4 | - | - |
| GPT-2L (16x SAE) | | | 3.5 | 1.8 | - | 7.4 | 4.2 | - |
| GPT-3L (16x SAE) | | | 3.2 | 2.3 | 1.1 | 9.6 | 5.0 | 4.5 |
| GPT-3.5-turbo | | | $\rho$ (**Random-only**) $(\%, \uparrow)$ | | | $\rho$ (**Top-and-Random**) $(\%, \uparrow)$ | | |
| Model | Size | Loss | Layer 1 | Layer 2 | Layer 3 | Layer 1 | Layer 2 | Layer 3 |
| CRATE-1L | 6.54M | 4.06 | 4.8 | - | - | 3.9 | - | - |
| CRATE-2L | 6.64M | 3.55 | 8.2 | 6.0 | - | 7.5 | 8.0 | - |
| CRATE-3L | 6.74M | 3.46 | 9.1 | 9.2 | 6.9 | 9.5 | 9.1 | 8.3 |
| GPT-1L | 6.64M | 3.83 | 9.0 | - | - | 9.0 | - | - |
| GPT-2L | 6.83M | 3.23 | 2.2 | 1.6 | - | 4.3 | 4.4 | - |
| GPT-3L | 7.03M | 3.11 | 3.0 | −0.3 | 1.2 | 7.0 | 3.1 | 3.0 |
| GPT-1L (16x SAE) | | | 2.6 | - | - | 4.7 | - | - |
| GPT-2L (16x SAE) | | | 3.4 | 1.6 | - | 5.0 | 2.9 | - |
| GPT-3L (16x SAE) | | | 2.8 | 1.8 | 1.2 | 7.4 | 3.8 | 3.2 |

**Simulation model.** The official implementation of the simulation model utilizes `text-davinci-003` (now named `gpt-3.5-turbo-instruct`), which no longer supports retrieving the logprobs through the API, so we use `LLaMA-2-70B` as an equally capable replacement (Touvron et al., 2023). For more accessible evaluations, we use `LLaMA-2-7B` instead. We show the difference in interpretability caused by different simulation model size on `LLaMA-2-7B` and `LLaMA-2-70B` in Table 7. Empirical results show that although `LLaMA-2-7B` has overall lower scores and higher variance than `LLaMA-2-70B`, it doesn't affect essential conclusions about the apple-to-apple comparison between CRATE and GPT.

Table 7: Interpretability measure of GPT and CRATE-GPT on the Pile dataset based on the OpenAI Top-and-random metric. Explanation model: `GPT-3.5-turbo`. Simulation model: `LLaMA-2-7B/LLaMA-2-70B`.

| | Interpretability (7B) (%, ↑) | | | Interpretability (70B) (%, ↑) | | |
|---|---|---|---|---|---|---|
| | Layer 1 | Layer 2 | Layer 3 | Layer 1 | Layer 2 | Layer 3 |
| CRATE-1L | 3.9 | - | - | 6.4 | - | - |
| CRATE-2L | 7.5 | 8.0 | - | 7.4 | 7.1 | - |
| CRATE-3L | 9.5 | 9.1 | 8.3 | 10.4 | 7.4 | 6.5 |
| GPT-1L | 9.0 | - | - | 13.4 | - | - |
| GPT-2L | 4.3 | 4.4 | - | 6.4 | 7.8 | - |
| GPT-3L | 7.0 | 3.1 | 3.0 | 10.1 | 3.2 | 6.3 |

| | | Layer 1 | Layer 2 | Layer 3 | Layer 4 | Layer 5 | Layer 6 |
|---|---|---|---|---|---|---|---|
| 7B | CRATE-6L | 10.5 | 7.4 | 5.8 | 8.0 | 8.1 | 5.7 |
| | GPT-6L | 13.7 | 5.2 | 1.9 | 0.6 | 5.6 | 6.9 |
| 70B | CRATE-6L | 10.1 | 6.5 | 7.0 | 8.3 | 9.4 | 0.7 |
| | GPT-6L | 14.5 | 6.2 | 2.5 | 0.7 | 3.9 | 4.1 |

## C ANALYSIS ON ACTIVATION SPARSITY

We demonstrate the activation sparsity of CRATE compared to GPT in Figure 9. We observe that the activations of CRATE are higher than GPT. One might have the confusion about why CRATE is designed to be sparse but the activations evaluated is denser than GPT. Note that the sparsity evaluated in standard transformer model is output from the hidden layer of the MLP layer, which is the activation matrix $A$ *before applying to the residual stream*, as shown in Figure 10. The actual representations in standard transformers, which are after applying the residual stream, are not sparse at all. In contrast, the sparsity evaluated in CRATE is the actual representations $A_t$ (*including the residual stream*).

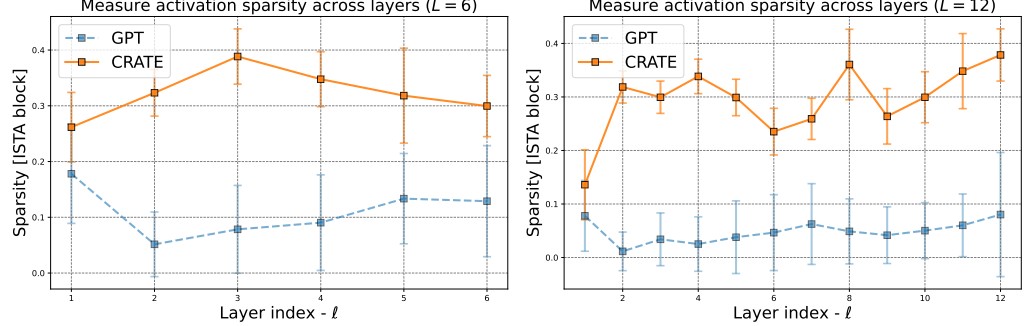

Figure 9: Layer-wise activation sparsity of CRATE and GPT. *Left:* 6L models. *Right:* 12L models.

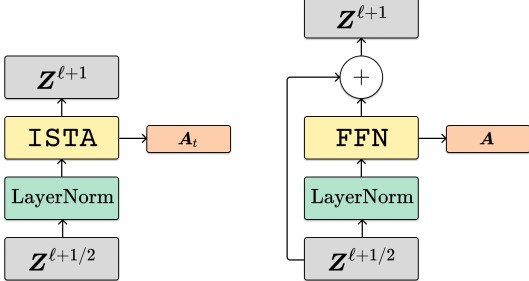

Figure 10: Extracting sparse code $A_t$ from CRATE and hidden layer output $A$ from GPT.

We also present the activation dynamics of CRATE and GPT with the progression of the pre-training process in Figure 11. We observe a strong trend that the sparsity of CRATE monotonically decreases in the early stage (trained on 1.6B tokens), which aligns with the design purpose. In the late stage (16B, 160B tokens), the sparsities in the early sites ($L < 12$) significantly decreases, which also aligns with the design purpose. On the other hand, GPT never appears to have a decreasing trend of activation sparsity over layers across the whole pre-training stage, indicating a systematic difference between the sparsity dynamics between CRATE and GPT. One counter-intuitive observation is that the decreasing trend fades as the stage moves on. Our hypothesis is that CRATE overfits on the next token prediction task due to the large amount of tokens trained.

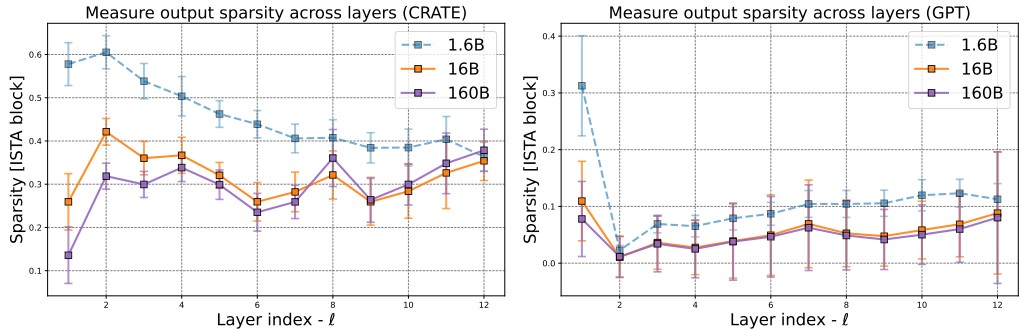

Figure 11: Layer-wise activation sparsity w.r.t. tokens trained. *Left:* CRATE language model. *Right:* GPT.

## D    DETAILS ON INTERPRETATION SCORE DISTRIBUTIONS

We visualize the distributions of layer-wise interpretation scores of CRATE and GPT with $L = 12$ in Figure 12. We exclude cases where activations sampled from GPT-Base (random-only metric) are all zeroes, as in these cases the correlation $\rho$ will be undefined. This results in a smaller number counted in the GPT activations in the first two rows.

## E    DETAILS ON SPARSE AUTO-ENCODER

### E.1    SPARSE AUTOENCODER AND DICTIONARY LEARNING

The dictionary learning model is an MLP with a single hidden layer. It is trained as an auto-encoder using input weights as the encoder that maps the input activations to a higher dimension, and output weights as the decoder. Formally, given activation $\boldsymbol{a} \in \mathbb{R}^h$ sampled from $\boldsymbol{A} \in \mathbb{R}^{h \times N}$, the encoder $\boldsymbol{W}_1, \boldsymbol{b}_1$ with dimension multiplicator $\mu$ maps the activations to a hidden representation $\boldsymbol{h} \in \mathbb{R}^{\mu h}$, whereas the decoder $\boldsymbol{W}_2, \boldsymbol{b}_2$ maps the representation back to the original dimension $\hat{\boldsymbol{a}} \in \mathbb{R}^h$. The dictionary learning objective can thus be expressed as

$$\bar{\boldsymbol{a}} = \boldsymbol{a} - \boldsymbol{b}_2 \tag{14}$$

$$\boldsymbol{h} = \mathrm{ReLU}(\boldsymbol{W}_1 \bar{x} + \boldsymbol{b}_1) \tag{15}$$

$$\hat{\boldsymbol{a}} = \boldsymbol{W}_2 \boldsymbol{h} + \boldsymbol{b}_2 \tag{16}$$

$$\mathcal{L} = \frac{1}{|\boldsymbol{A}|} \sum_{\boldsymbol{a} \in \boldsymbol{A}} \|\boldsymbol{a} - \hat{\boldsymbol{a}}\|_2^2 + \lambda \|\boldsymbol{h}\|_1 \tag{17}$$

### E.2    DETAILED SETUP

We train the sparse auto-encoders on the train split of the uncopyrighted Pile dataset until convergence. Following Bricken et al. (2023) and Conmy (2023), we adopt the resampling strategy to re-train the dead features, and the learning rate scheduling strategy to improve recovery rate. For

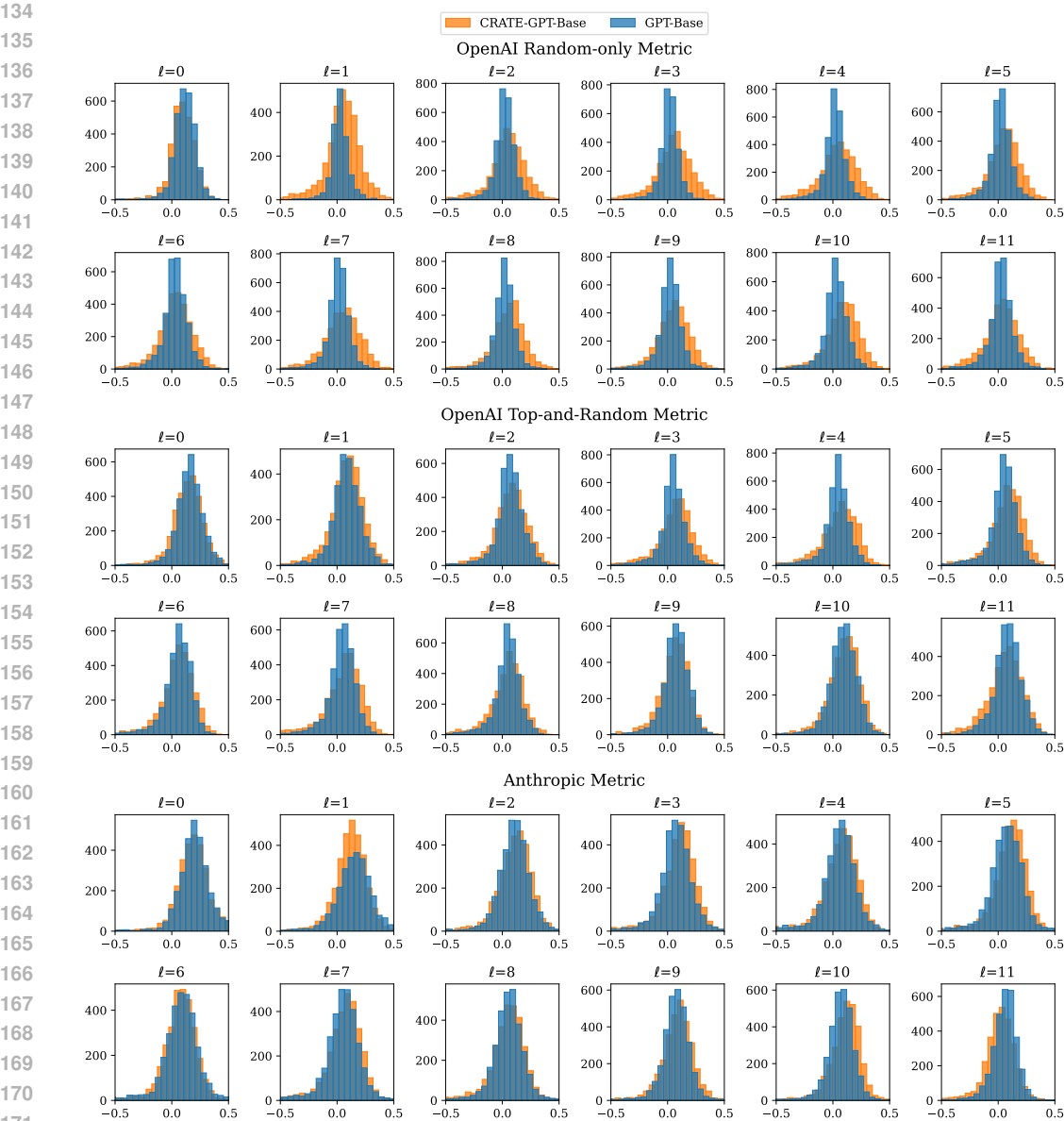

Figure 12: Distribution of the interpretation scores over CRATE-12L and GPT-12L. $x$-axis: interpretation score. $y$-axis: count of neurons falling in the corresponding interval of interpretation score.

implementation, we mainly follow Conmy (2024), with $\lambda_{\ell_1} = 1.6 \times 10^{-4}, \alpha = 1.2 \times 10^{-3}$ for all sizes of models. We evaluate using the average loss of randomly sampled batches on the validation split of the uncopyrighted Pile dataset.

### E.3    LOSS CURVE

The loss curves of training the sparse auto-encoders are shown in Figure 13. Generally, resampling boosts the performance of the recovery score, which aligns with the conclusions shown in Bricken et al. (2023) and Conmy (2023). We also observe an increasing trend of performance with the increases of the SAE multiplication factor $\mu$ and model size $L$.

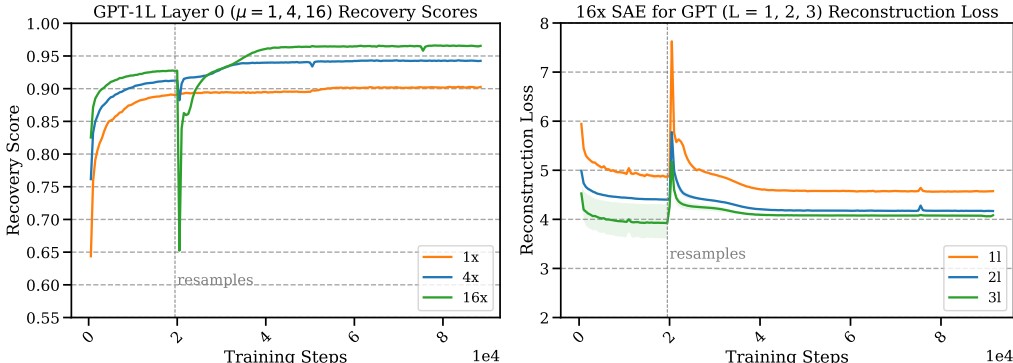

Figure 13: *Left:* The recovery scores of GPT-1L ($\ell = 0$) with SAE multiplication factors $\mu = \in \{1, 4, 16\}$. *Right:* The reconstruction loss of SAE with $\mu = 16$ on different sizes of GPT models $L \in \{1, 2, 3\}$, averaged across all layers.

### E.4 PERFORMANCE

The performance of sparse auto-encoders of CRATE-GPT and GPT under a variety of settings (model $L$, $\ell$ and sparse autoencoder width multiplication factor $\mu$) are shown in Table 8. The percentages of dead neurons for all layers of $L \in \{1, 2, 3\}$ are less than 1%.

Table 8: Reconstruction loss and recovery score of the sparse autoencoders on CRATE and GPT.

| $\mu = 16$ | Size | Loss | **Reconstruction Loss ($\downarrow$)** Layer 0 | Layer 1 | Layer 2 | **Recovery Score ($\%, \uparrow$)** Layer 0 | Layer 1 | Layer 2 |
|---|---|---|---|---|---|---|---|---|
| GPT-1L | 6.64M | 3.83 | 4.35 | - | - | 95.0 | - | - |
| GPT-2L | 6.83M | 3.23 | 3.50 | 3.45 | - | 95.2 | 92.2 | - |
| GPT-3L | 7.03M | 3.11 | 3.38 | 3.39 | 3.29 | 94.6 | 94.8 | 92.4 |
| CRATE-1L | 6.54M | 4.06 | 4.33 | - | - | 93.6 | - | - |
| CRATE-2L | 6.64M | 3.55 | 4.12 | 3.80 | - | 95.7 | 95.7 | - |
| CRATE-3L | 6.74M | 3.46 | 4.05 | 3.99 | 3.77 | 93.0 | 93.9 | 95.0 |

| $\mu = 4$ | Size | Loss | **Reconstruction Loss ($\downarrow$)** Layer 0 | Layer 1 | Layer 2 | **Recovery Score ($\%, \uparrow$)** Layer 0 | Layer 1 | Layer 2 |
|---|---|---|---|---|---|---|---|---|
| GPT-1L | 6.64M | 3.83 | 4.34 | - | - | 93.7 | - | - |
| GPT-2L | 6.83M | 3.23 | 3.59 | 3.56 | - | 92.7 | 88.6 | - |
| GPT-3L | 7.03M | 3.11 | 3.45 | 3.50 | 3.34 | 92.2 | 94.9 | 89.9 |
| CRATE-1L | 6.54M | 4.06 | 4.39 | - | - | 92.1 | - | - |
| CRATE-2L | 6.64M | 3.55 | 4.37 | 3.93 | - | 93.7 | 93.8 | - |
| CRATE-3L | 6.74M | 3.46 | 4.03 | 4.11 | 3.81 | 92.6 | 92.6 | 92.6 |

| $\mu = 1$ | Size | Loss | **Reconstruction Loss ($\downarrow$)** Layer 0 | Layer 1 | Layer 2 | **Recovery Score ($\%, \uparrow$)** Layer 0 | Layer 1 | Layer 2 |
|---|---|---|---|---|---|---|---|---|
| GPT-1L | 6.64M | 3.83 | 4.93 | - | - | 95.0 | - | - |
| GPT-2L | 6.83M | 3.23 | 3.89 | 3.75 | - | 89.0 | 82.2 | - |
| GPT-3L | 7.03M | 3.11 | 3.63 | 3.61 | 3.58 | 86.9 | 91.0 | 84.9 |
| CRATE-1L | 6.54M | 4.06 | 4.69 | - | - | 86.2 | - | - |
| CRATE-2L | 6.64M | 3.55 | 4.68 | 4.29 | - | 90.4 | 88.9 | - |
| CRATE-3L | 6.74M | 3.46 | 4.39 | 4.38 | 4.16 | 97.0 | 89.2 | 88.1 |

### E.5 INTERPRETABILITY

**Does CRATE have more optimal representation than GPT in terms of interpretability?** As it's hard to decide how much interpretability gain it is from CRATE to CRATE-SAE directly (as explained in Section 5.3), we compare the interpretability *improvement* of CRATE-SAE over CRATE to the interpretability improvement of GPT-SAE over GPT.

The interpretability of GPT-SAE is already included in Figure 4. The interpretability of CRATE-SAE under the OpenAI TaR and Anthropic metrics are shown in Table 9.

Table 9: Interpretability of CRATE-SAE under the OpenAI TaR and Anthropic metics.

| | OpenAI TaR (↑) | | | Anthropic (↑) | | |
|---|---|---|---|---|---|---|
| | Layer 1 | Layer 2 | Layer 3 | Layer 1 | Layer 2 | Layer 3 |
| 1L | 6.0 | - | - | 17.9 | - | - |
| 2L | 7.7 | 5.2 | - | 21.7 | 12.4 | - |
| 3L | 7.2 | 6.4 | 4.6 | 19.3 | 18.4 | 11.6 |

### E.6 STEERING THE LM OR SAE

In comparison to post-hoc trained SAEs, built-in sparsification processes, such as the one we proposed in this paper, have the potential to be steered with perfect fidelity. As visualized in Figure 14, post-hoc approaches like SAE require steering the model with the decomposed hidden states $h$, whose encoding and decoding processes are both *lossy*. An imperfect reconstruction systematically leads to *distortions* of the steering signal upon the hidden states, and thus affects downstream applications of the GPT-SAE model. In contrast, CRATE doesn't include any approximation that distorts the steering signal, so the signal can be propagated without loss of fidelity. This conclusion does not change whether the performance of GPT-SAE outperforms CRATE or not.

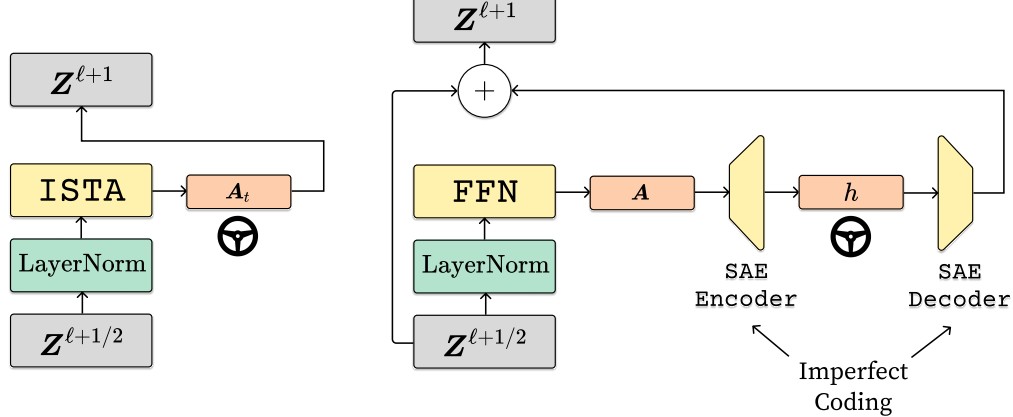

Figure 14: Illustration of steering a language model directly or using SAE.

## F FURTHER QUALITATIVE RESULTS ON INTERPRETABLE NEURONS

This section lists some further qualitative examples of tokens and their activations when $L = 3$, including examples of the GPT-SAE activations. Specifically, we demonstrate two neurons in each model. Tokens with a deeper blue background have a higher activation. Explanations $k_i$ and scores $s_i$ are obtained by Algorithm 1.

### F.1 CRATE-3L, LAYER 0, NEURON 288

**OpenAI Evaluation** **Score:** 0.44478400135318646
**Explanation:** information related to the regulation of mRNA expression and its role in carbohydrate metabolism, with a focus on CRC cells and gene signaling in the context of cancer development.
**Top Activations**

```
animal models of cartilage degradation ([@b 41 - mm r - 16 - 04 - 38 41 ]). Among
these cytok ines , IL - 1 β is highly overe xp ressed in the cart ilage and in
the syn ov ial tissue , while the expression of IL - 1 receptor antagonist ([
@ b 42 - mm r -
```

raft s . Moreover , AC OX 1 overe xp ression atten uated the aug mentation of migration and invasion of CRC cells by mi R – 15 b – 5 p overe xp ression . In conclusion , our study demonstrated a functional role of the S IRT 1 / mi R – 15 b – 5 p / AC OX 1 axis

**Random Activations**

ed with Peter Braun , the Mor av ian \n mission ary in Ant ig ua ; and to that correspondence he owed in part his \n interest in missionary work . But that was not the end of the Bre thren ' s \n inf luence . At all meetings addressed by the founders of the proposed \n Soc iety , the speaker repeatedly

ESA ) and the American Association for the Advance ment of Science ( AA AS ) have well – developed and successful science policy fellows hips . These programs acknowledge that scientists can play important roles in directing new laws and policies in their field , and that their expertise is needed for effective decision – making \ [[ @ B 81 – in

**Anthropic Evaluation    Score:** 0.2813215497394413
**Explanation:** Phrases related to molecular biology and gene expression, specifically in the context of mRNA transcription and its activation or inhibition. Additionally, there are mentions of certain proteins (IPOTENT, cytokine, IRT), cellular processes (proliferation,

**Top Activations**

b – 5 p overe xp ression .

cases showed EG FR overe xp ression .

, TL R 2 overe xp ression in

IL – 1 $\beta$ mRNA expression in the

**Random Activations**

G . al bid us * T MW

had suffered from heat contact ur tic aria

" al ive ": true , \n

= _ mm _ pack us _ ep

## F.2   GPT-3L, Layer 2, Neuron 289

**OpenAI Evaluation    Score:** -0.3236237946813878
**Explanation:** The provided text contains multiple sections, but the activations given for Neuron 4 seem to be related to genetic and statistical data (e.g., population, CI, percent, risk association, and recessive models). Given this, the main thing this neuron does is identify

**Top Activations**

in Asian population . Similarly , in Caucasian population , the rs 499 776 polymorph ism attributes risk association in hom ozyg ote OR 0 . 70 ( 95 % CI [ 0 . 50 – 0 . 98 ]), dominant OR 3 . 57 ( 95 % CI [ 2 . 34 – 5 . 27 ]), and recess ive models OR 0

* SE * = 0 . 04 40 , * t * = – 1 . 07 75 , * p * \ > 0 . 05 , 95 % CI (– 0 . 13 38 , 0 . 0 390 ) for the Slov ak ian villagers story \ ], therefore indicating full mediation by exoner ations and out – group focused emotions

**Random Activations**

long er the vortex , it ' s the smooth current of rotating air which is next to \n the vortex , and we use the upd raft of this air ." Taking advantage of the free \n lift in this upd raft of air is called " wake – energy retrieval ." ... on long – \n haul flights , fuel savings of between

ushing . \n \n          S igh called her supervisor . Sergeant Sweeney and Deputy Ray responded \n \n and moved Don ery so that S igh could search his cell . Don ery had been in his new \n \n cell for less than five minutes when the toilet overfl owed and water began flowing out \n \n of

**Anthropic Evaluation    Score:** 0.06532542667915378

**Explanation:** strings containing specific numbers and alphanumeric characters, such as "CI-50-.", "e-44-", "87-", and "f-". Additionally, it activates slightly for certain words like "cost", "weeks", "disability", and

**Top Activations**

| 95 % CI [ 0 . 50 – |
|---|

| 95 % CI , 0 . 13 – |
|---|

| f \ " ], [ 0 . 22 22 |
|---|

| 95 % CI [ 0 . 50 – |
|---|

**Random Activations**

| { 8 }{ 45 } \ pi \\ \n |
|---|

| instead of that silly website . <\|endoftext\|> How |
|---|

| is free software ; you can redist ribute |
|---|

### F.3    GPT-3L-SAE-16X, LAYER 2, NEURON 57

**OpenAI Evaluation    Score:** 0.1427145260798203

**Explanation:** months or the word "Bank" followed by a year.

**Top Activations**

| No . 18 – 20 609                                February 21 , 2020 \n |
|---|

| with the compact – open top ology , is a locally compact group .' \n author : \n – ' Nic olas Rad u [ ^ 1 ]' \n date : ' July 15 , 2016 ' \n title : \| \n     A top ological characterization of the M ouf ang \ \n    property for compact polyg ons |
|---|

**Random Activations**

| of DN * db / db * mice . \ \n ( ** A ** ) Ur inary album in to creat in ine ratio . ( ** B ** ) Ser um u rea nitrogen . ( ** C ** ) Left kidney weight to body weight ratio . ( ** D ** ) HE st aining . Bar    = |
|---|

| this email : ot isd ark o 60 @ yahoo . com \n \n HE FIX THE FO LLOW ING PR OB LE MS TO ALL \n \n AC R OSS THE GL OB E ON : \n \n 1 . Getting your lover or husband back \n \n 2 . Spiritual bullet proof \n \n 3 . Training \n \n 4 . Money |
|---|

**Anthropic Evaluation    Score:** 0.2540425061001668

**Explanation:** dates and specifically, the month and day for a given year. The neuron is not activated by the year alone, and it requires both the month and day for a complete activation.

**Top Activations**

| February 21 , 2020 |
|---|

| \n date : ' July 15 , 2016 |
|---|

| field , Missouri ( December 15 , 2014 |
|---|

| February 5 , 1998 |
|---|

**Random Activations**

| \n   Can ola oil |
|---|

| } \n  \n . c ke |
|---|

| uana when a draw would have clin ched |
|---|

| . </ p > \n \t |
|---|

