# OpenReview forum: "Improving Neuron-level Interpretability with White-box Language Models"
_ICLR.cc/2025/Conference — ICLR 2025 Conference Withdrawn Submission_

### Official Review · Reviewer_1u9C · 2024-11-04

**Soundness:** 3
**Presentation:** 3
**Contribution:** 2
**Rating:** 5
**Confidence:** 3

**Summary:**

The paper introduces the CRATE architecture for language modeling, aimed at enhancing interpretability of language models, particularly at the neuron level. The architecture integrates a sparse coding mechanism in its ISTA block (an alternative to the MLP block in GPT models), which allows for improved sparsity and interpretability. However, the model’s language pretraining performance is suboptimal compared to GPT by a noticeable margin. The CRATE model is evaluated on interpretability benchmarks, showing improvements in neuron activation consistency on semantics. But the ablation on the architecture design choices are rather insufficient.

**Strengths:**

1. The CRATE architecture, integrated with the sparse coding mechanism in the ISTA block, has better neuron-level interpretability than GPT. Individual neurons in CRATE respond more consistently to specific semantic concepts, making the correspondence between neuron activations to language features purer than those in GPT neurons.
2. The model got high scores in neuron-level probing and outperforms GPT in neuron interpretability metrics, suggesting that CRATE’s architectural design aligns well with interpretability goals.

**Weaknesses:**

1. While CRATE improves interpretability, it performs unfavorably in language pretraining tasks as shown in Fig 2, indicating a possible trade-off between architectural interpretability and language modeling performance.
2. When interpreted using SAE, CRATE model underperforms in OpenAI metrics, implying that CRATE’s interpretability gains are not consistent across all interpretability methods.
3. The ISTA block’s thresholded activation approach might be over-engineered. The ISTA block resembles a MLP block with two layers, but presented with more complex formulations . Even though the paper (and the CRATE paper it follows) suggest that this design is derived from theoretical principles, there is no ablations in this paper to discuss the trade-offs of using such design, compared to other alternatives. I am curious whether a simpler sparsity constraint on intermediate MLP activations could potentially achieve better balance between interpretability and performance.
4. The paper discussed that improved interpretability may not come from a performance gap. However, the more important question should be if the gap is unavoidable due to the sparsity imposed on the ISTA block, which is essential for the interpretability improvement. One can train the CRATE model really bad to achieve both bad interpretability and performance, but that does not prove that there is no negative correlation between interpretable architecture design and language pre-training performance.

**Questions:**

1.  Given that ISTA essentially operates like a ReLU with thresholded features, would a simpler sparsity constraint on MLP activations achieve similar results?
2. If I did not misunderstood, the CRATE model is trained in a similar approach to common GPT models. Although the paper derived all the components (ISTA and MSSA blocks) via the rate reduction principle, there is no absolute guarantee that the model acts according to this principle and may behave unexpectedly. In fact, I think the separation of compression and sparsification feels somewhat artificial when discussing deep models trained with backpropagation. Could you clarify why this architecture is still considered white-box?

---

### Official Review · Reviewer_45EJ · 2024-11-04

**Soundness:** 3
**Presentation:** 2
**Contribution:** 2
**Rating:** 5
**Confidence:** 3

**Summary:**

In this study, the Coding RAte TransformEr (CRATE) architecture is adapted to the task of autoregressive language modeling for the first time. CRATE integrates sparse coding directly into the model architecture for better interpretability, moving beyond post-hoc methods. While it performs worse than a vanilla GPT-2 baseline in terms of language modeling perplexity, CRATE shows improved interpretability of neurons as compared to GPT-2.

**Strengths:**

The CRATE algorithm places the work on a strong mathematical basis. The simulation scoring evaluation shows that the CRATE architecture is quantifiably more interpretable than a naive baseline.

**Weaknesses:**

The reported increase in interpretability for the largest, 12-layer model is fairly modest (16.3%). Mean interpretability scores seem low in absolute terms across the board (usually less than 10 out of 100), indicating that CRATE is only slightly less black box than a normal language model, rather than a genuinely "white box" architecture, as the title suggests. The examples in Figure 1 seem to be cherrypicked to have much higher than average interpretability scores (in the 36 to 50 range). Indeed, because Table 2 shows that CRATE has lower variance in interpretability scores than GPT-2, the examples shown in Fig. 1 must be several standard deviations more interpretable than the mean.

The abstract says "we introduce a white-box transformer-like architecture named Coding RAte TransformEr (CRATE)," but this is misleading since the body of the text recognizes that CRATE was previously introduced by Yu et al. (2023), and the main contribution of the present work is to build a language model on top of the CRATE representation learning framework. The abstract should be updated to more accurately reflect the novel contribution of this work.

**Questions:**

The loss curve for CRATE in Figure 2 qualitatively looks like it has not plateaued. Do the authors expect that further training could cause CRATE to catch up to the transformer baseline in terms of perplexity? How do the loss curves look for model sizes other than Base? It would be quite compelling if, at a smaller model size trained on more tokens, CRATE was found to catch up with the transformer.

---

### Official Review · Reviewer_C7cn · 2024-11-06

**Soundness:** 1
**Presentation:** 3
**Contribution:** 2
**Rating:** 3
**Confidence:** 3

**Summary:**

This paper proposes CRATE language models, a new alternative architecture to transformers that aims to be more inherently interpretable by encouraging a sparse/disentangled representation. This architecture was introduced for vision models in a recent paper and this paper adapts it to the vision setting. They show it increases interpretability of individual neurons in automated evaluations.

**Strengths:**

Addresses an important problem. Some promising results, such as increased interpretability of individual neurons over standard transformers when measured with automated interpretability. Mostly well written. Overall I think the method has promise but is currently lacking evidence to really show whether it's a meaningful improvement over previous methods.

**Weaknesses:**

Overall I feel like the paper is overclaiming a bit in many places:
- To begin with, I really don't like the name white-box, as that implies the model is easy to understand. While it is an improvement over standard transformer, we are far from fully understanding the model, and I think a more accurate name would be dark-grey-box or something. I think calling this an interpretable-by-design model instead of white-box would be better.
- I disagree with some of the authors claims regarding limitations of SAE models, in particular claims that they scale poorly are not supported by the references provided
    - Templeton et al. cited on line 78 actually shows how to successfully scales SAEs to frontier language models
    - citation in line 124 simply shows later layers of the same attention sae are less interpretable, which is not related to scaling. Also don’t think this is true for saes trained on MLP neurons or residual stream.
    - Overall seems like sparse-autoencoders scale better than the proposed models, i.e. people have trained saes on much bigger models than gpt-2 without issues, and the computational cost of training an SAE is smaller than the cost of training a new language model.
- The paper claims to solve the issue of introducing reconstruction loss when training SAEs, but the introduced method has worse language modeling capabilities than standard models, and the scale of this difference seems similar to that introduced by SAEs


I'm not convinced these models are much more interpretable than previous methods:
- The shown examples show good explanation for the highly activating input the random activations seem not related to explanation at all
- Are the example neurons cherry-picked? I would like to examples of randomly selected neurons
- The paper would benefit for an additional interpretability study with human evaluators
- Doesn't look like this method is significantly more interpretable than original GPT neurons on larger model i.e. L=12. Only random-only score is improved, but it is very low to begin with so I'm not sure how meaningful and improvement there is. Most metrics seem to be within the error margin (why are error bars scaled down 10x? feels intentionally misleading).

Some parts hard to follow, in particular section 3:
- I feel like section 3 does not describe the architecture in sufficient detail. For example, what is the coding rate R? I should be able to follow the main method without reading the references.

Minor:
- Eq. 15 should have \bar{a} instead of \bar{x]

**Questions:**

- How exactly is the model trained to predict next token? I don’t see any task related loss terms discussed.
- Why do you discard the last layer for interpretability evaluation? Seems a little unfair to me. Don’t we also want last layer to be interpretable?
- Algorithm 1 says you are measuring correlation. Why is the interpretation score not between [-1, 1]?
- How many neurons were used for evaluation in figure 4?
- Are the fig 4 error bars variance or standard deviation?
- Why is the variance zero for models with few layers and then explodes for bigger ones?
- Table 2 says variance of interpretability scores is much smaller for 12L crate models than 12L gpt models, however in fig 12 it looks like the opposite. Why is this the case?
- What is the sparsity i.e. average L0 of the trained SAE models in Table 8?
- What expansion factor mu was used for SAEs in fig 4?
- How does the training time compare to training a GPT-model of similar size?
- The methods mostly improves random-only scores. Could this be because the neurons are less sparse?

---

### Note · Authors · 2024-11-19

I have read and agree with the venue's withdrawal policy on behalf of myself and my co-authors.